# Local hippocampal fast gamma rhythms precede brain-wide hyperemic patterns during spontaneous rodent REM sleep

Antoine Bergel [1,2,3], Thomas Deffieux[2], Charlie Demené[2], Mickaël Tanter[2] & Ivan Cohen[1]

Rapid eye movement sleep (REMS) is a peculiar brain state combining the behavioral components of sleep and the electrophysiological profiles of wake. After decades of research our understanding of REMS still is precluded by the difficulty to observe its spontaneous dynamics and the lack of multimodal recording approaches to build comprehensive datasets. We used functional ultrasound (fUS) imaging concurrently with extracellular recordings of local field potentials (LFP) to reveal brain-wide spatiotemporal hemodynamics of single REMS episodes. We demonstrate for the first time the close association between global hyperemic events – largely outmatching wake levels in most brain regions – and local hippocampal theta (6–10 Hz) and fast gamma (80–110 Hz) events in the CA1 region. In particular, the power of fast gamma oscillations strongly correlated with the amplitude of subsequent vascular events. Our findings challenge our current understanding of neurovascular coupling and question the evolutionary benefit of such energy-demanding patterns in REMS function.

[1] Sorbonne Université, CNRS, INSERM, Institut de Biologie Paris Seine-Neuroscience, 9 quai Saint-Bernard, 75005 Paris, France. [2] Institut Langevin, ESPCI ParisTech, PSL Research University, CNRS UMR7587, INSERM U979, 17 rue Moreau, 75012 Paris, France. [3] Université Paris Diderot, Sorbonne Paris Cité, 7 rue Thomas Mann, 75013 Paris, France. These authors contributed equally: Mickaël Tanter, Ivan Cohen. Correspondence and requests for materials should be addressed to A.B. (email: antoine.bergel@espci.fr) or to I.C. (email: ivan.cohen@inserm.fr)

The memory function and physiological mechanisms of sleep in animal models have been highlighted over the last decade by techniques that enable the selective suppression and manipulation of brain oscillations[1,2], event-triggered imaging[3] and the activation of targeted neuronal ensembles[4]. In particular, previous work on hippocampal replay in rodents have established that sharp-wave ripples support memory consolidation processes by facilitating the transfer of newly formed memory traces in the hippocampus towards stable cortical sites during sleep[5,6] and that they are critical for spatial learning during the awake state[7]. Most of these breakthroughs have focused on hippocampal recordings during slow-wave sleep (SWS), leaving rapid-eye-movement sleep (REMS) comparatively understudied and confining its putative role to dreaming and emotional processing[8]. However, recent evidence shows that REMS plays a major role in hippocampal plasticity[9], that theta oscillations are critical for hippocampus-dependent memory consolidation in mice[10] and that REMS allows selective pruning and consolidation of newly formed synapses throughout development and learning[11]. These new findings strongly question the functional dissociation between selective processes occurring during SWS and REMS.

REMS is a peculiar brain state characterized by wake-like electrophysiological patterns accompanied by the behavioral components of sleep[12,13]. The neurophysiology of REMS includes prominent features that strongly differ from wake such as high cholinergic tone in the cortex, muscle paralysis, irregular heart rate, irregular breathing, rapid-eye movements, and penile erection[14,15]. However, it has been very challenging to differentiate between wake and REMS based solely on local field potentials (LFP) and without monitoring the discharge patterns of principal neurons[16], which has prompted the need for multimodal recording approaches to build a comprehensive model of REMS. As with SWS, REMS is not divided into stages but rather described in terms of a tonic component (including high arousal threshold, low-amplitude synchronized electroencephalogram (EEG), and muscle atonia) that is interleaved with phasic events (including bursts of oculomotor activity, irregular breathing and prominent electrographic events such as ponto-geniculo-occipital waves, theta, and gamma bursts)[17–20].

Due to the transient nature of REMS and to the significant caveats associated with actual recording techniques, a simultaneous monitoring of electrographic and vascular events during spontaneous sleep has been very hard to achieve. This results in an important lack of data regarding the activity of distributed networks during REMS and a poor understanding of the physiological function of its phasic events. Previous studies have reported that REMS is associated with increased cerebral blood flow in cats[21], surges in heart rate and arterial pressure in rats[22], and functional microstates in humans[23,24]. Importantly, rich oscillatory content precedes vascular events, suggesting that they may act in inter-area communication to synchronize distributed networks[18]. However, both the physiological function of REMS and its phasic/tonic components remain elusive.

This study imaged brain activity during natural REMS in rats using a combination of electrophysiology and fUS imaging[25]. This novel neuroimaging modality based on ultrafast ultrasound imaging[26] provides new insights into brain dynamics thanks to comprehensive imaging of hemodynamics in conjunction with local recordings of electrographic activity. In contrast with EEG-functional Magnetic Resonance Imaging (fMRI), which entails habituation, care with electrode design and sleep deprivation protocols, fUS-EEG appeared particularly well-suited for animal sleep studies because it unveiled the vascular correlates of selective brain rhythms over large distributed networks including deep structures and whole-brain hemodynamics at the sub-second scale[27].

We report imaging of brain-wide vascular hyperactivity specific to REMS in rats that previously performed a track-running task. We show that vascular activity also divides into tonic and phasic regimes, the latter exhibiting transient brain-wide hyperemic patterns, which we called vascular surges (VS). VS amplitude was strongest in the dorsal hippocampus and outmatched wake levels in nearly all brain regions, occasionally reaching up to a 100% increase in the cortical and hippocampal regions compared to a quiet wake state. We isolated precursors to VS in the theta (6–10 Hz) and high-gamma (70–110 Hz) bands of hippocampal LFP, and the intensity of each individual VS was best accounted for by the power of fast gamma, suggesting a strong association between local electrographic events and massive brain-wide vascular patterns.

## Results

**REMS is characterized by massive brain-wide hyperemia.** Using a traditional sleep scoring procedure based on movement detection (video tracking and accelerometer), neck electromyogram (EMG) and intra-hippocampal local field potentials (LFP), we were able to distinguish between four different states: quiet wake (QW), active wake (AW), non-REM sleep (NREMS), and REM sleep (REMS) [Fig. 1a]. Although the LFP patterns were strongly similar between AW and REM, as characterized by prominent peaks in the theta (6–10 Hz) and high-gamma (100–150 Hz) bands, low ripple content and high theta/delta ratio[28] [Supplementary Fig. 1], we found that cerebral blood volume (CBV) profiles strongly differ between these two states, whereas QW and NREMS showed relatively low CBV levels (bottom black line, Fig. 1a). REMS was characterized by elevated CBV levels for the whole duration of the REM episode in all recorded brain regions (more moderately in ventral thalamus and hypothalamus) compared with a baseline in the first 3 min of QW in each recording. This vascular activity during REMS fluctuated between moderate CBV levels matching AW levels (REM-TONIC) and massive CBV spikes (REM-PHASIC), which we called vascular surges (VS, red shaded boxes, Fig. 1b) and were present in nearly all brain regions but more prominent and stronger in all sub-regions of the dorsal hippocampus [Fig. 1b]. Thorough inter-individual analysis of VS amplitude across brain regions is provided later in the paper. These VS events, which we detected using a population activity threshold [Supplementary Fig. 2], typically lasted 5 to 30 s but could extend to over a minute in cortical regions. Typical transitions between NREMS and REMS episodes are shown in supplementary content in three animals over three different recording planes [Supplementary Movies 1–3].

We investigated whether the profile of the VS events was similar among animals. We analyzed from five animals for which we could reliably perform statistics (two animals were excluded because we could record <5 VS events overall) and computed the number of VS events, their duration, intensity, and activity ratio [Table 1]. The intensity of a given VS event was simply computed by spatially averaging the CBV signal in all brain voxels. The activity ratio is the proportion of "active pixels" as defined previously [Supplementary Fig. 2] to detect the VS events. From these temporal variables we could extract a mean and maximum value for each VS event. The profile of VS events was remarkably constant over all five animals (around four to six VS events per REM episode) and though the duration of VS events displayed high variability this was observed in four out of five animals. On the other hand, the intensity was roughly constant (from 23.2 to 34.8%) and activity ratio of VS events was very well preserved across animals (from 59.9 to 65.0%).

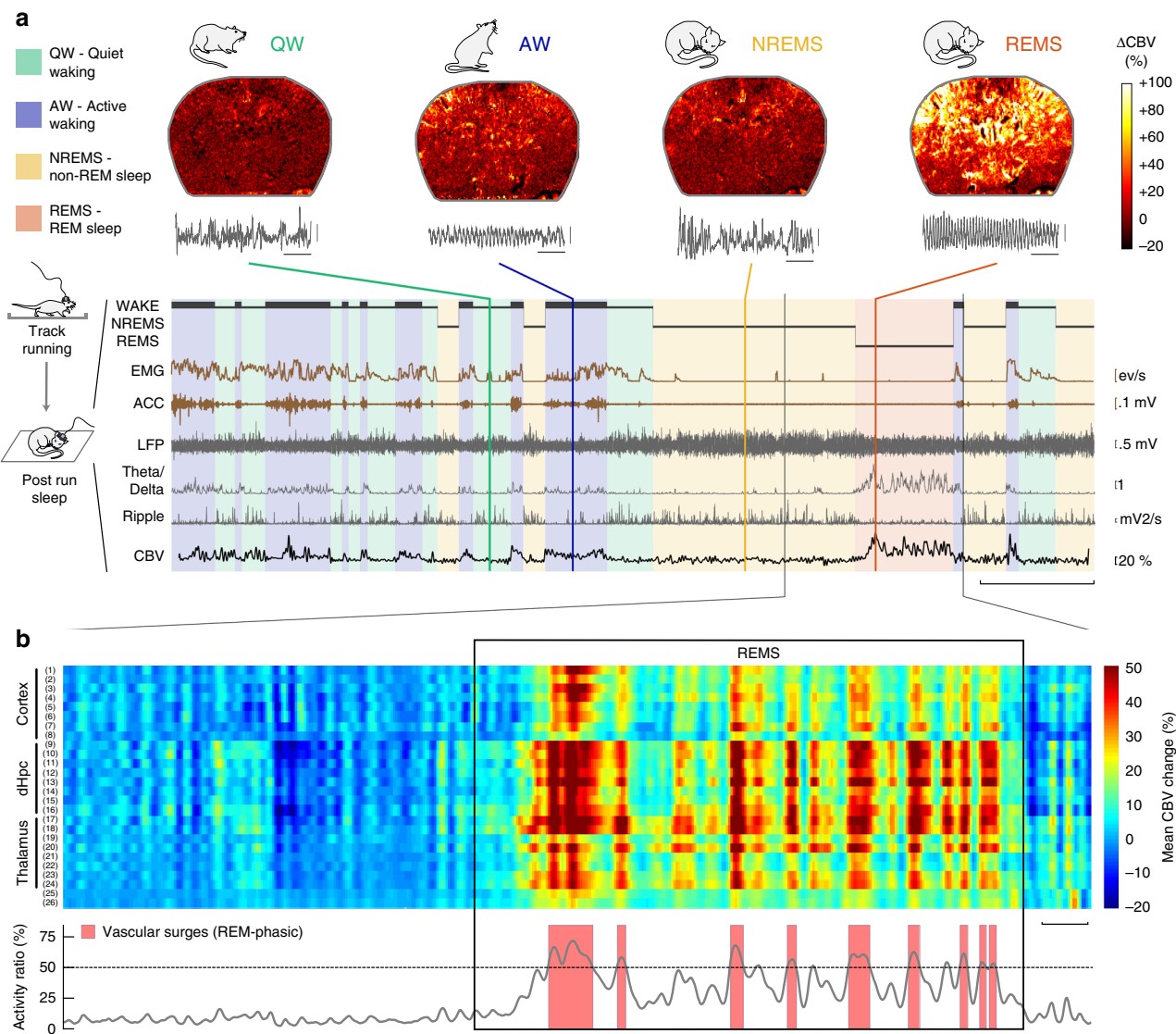

**Fig. 1** REM sleep is characterized by massive brain-wide hyperemia. **a** Typical fUS-EEG recording during sleep subsequent to a track-running task. Sleep scoring is performed by using neck electromyogram (EMG), head acceleration (ACC), and hippocampal local field potential (LFP) to discriminate between quiet wake (QW), active wake (AW), non-REM sleep (NREMS), and REM sleep (REMS). REMS periods are characterized by high theta/delta ratio, low ripple content and elevated cerebral blood volume (CBV). Scale bar: 5 min (Top) Four typical Power Doppler images (CBV, voxel resolution: $100 \times 100 \times 400\ \mu m$) for each state and corresponding 5-s LFP trace. Scale bar: 1 s. Note the similarity in the EEG patterns (theta activity) between AW and REMS, contrasted by the discrepancy in the vascular patterns. **b** Regional hemodynamics during a transition between NREMS to REMS. (Top) High-amplitude vascular patterns are present in almost all brain regions and more sustained in the dorsal thalamus and hippocampus. (1–2) Auditory cortex, left and right, respectively; (3–4) Primary somatosensory barrel field cortex; (5–6) Lateral parietal association cortex; (7–8) Retrosplenial cortex; (9–10) CA1 region; (11–12) CA2 region; (13–14) CA3 region; (15–16) Dentate gyrus; (17–18) Dorsal thalamus; (19–20) Post-thalamic nuclear group; (21–22) Ventral posteromedial thalamic nucleus; (23–24) Whole thalamus; (25–26) Hypothalamic region. Scale bar: 20 s. (Bottom) Phasic vascular events (vascular surges) are detected when the proportion of active voxels (gray curve) crosses 50% of total voxel amount. A pixel is «active» when its value is higher than one standard-deviation above active wake levels

**REMS hyperemic events are strongest in the dorsal hippocampus**. We quantified these different vascular regimes across brain regions and individuals using an MRI-based atlas-registration algorithm [Supplementary Fig. 3]. As expected, QW and NREM vascular regimes were relatively similar to baseline, except in the thalamus, which was hypoactive during NREM (Cohen's $d = -0.24$, $p < 10^{-3}$, two-tailed Mann–Whitney test). AW was significantly higher than QW (Cohen's $d = 1.06$, $p < 10^{-3}$, two-tailed Mann–Whitney test), especially in the cortical regions, while REM (and REM-phasic) strongly outmatched baseline levels, doing so more strongly in hippocampal regions (Cohen's

$d = 1.77$ and $3.87$, $p < 10^{-3}$, two-tailed Mann–Whitney test) [Fig. 2a and Supplementary Fig. 4]. To demonstrate that the CBV differences observed here are not spuriously arising from inter-individual averaging, we computed confidence intervals for each behavioral state for each of the five animals for which we could reliably perform averaging [Table 2]. This analysis confirmed that dorsal hippocampus and dorsal thalamus were strongly activated during REMS in all animals (Dorsal Hippocampus: from 19.46 to 29.21%; Dorsal Thalamus: from 17.23 to 24.11%; Cortex: from 10.02 to 20.78%), that this effect was even stronger during REM-PHASIC periods (Dorsal Hippocampus: from 39.99 to 49.78 %;

**Table 1 Variability of vascular surges (VS) events across animals**

| Animal | # REM episodes | # VS events | Duration (s) | Mean intensity (%) | Max intensity (%) | Mean activity ratio (%) | Maximal activity ratio (%) |
|---|---|---|---|---|---|---|---|
| Rat 1 | 6 | 28 | 9.67 ± 7.87 | 25.01 ± 3.88 | 31.54 ± 8.32 | 61.2 ± 5.9 | 68.4 ± 9.5 |
| Rat 2 | 6 | 26 | 13.27 ± 16.00 | 25.50 ± 3.69 | 31.72 ± 8.58 | 61.9 ± 5.6 | 68.8 ± 8. 7 |
| Rat 3 | 8 | 25 | 5.12 ± 1.64 | 34.82 ± 6.00 | 41.05 ± 8.53 | 59.9 ± 3.1 | 66.8 ± 4.7 |
| Rat 4 | 6 | 24 | 8.24 ± 7.59 | 23.16 ± 5.00 | 29.12 ± 8.19 | 62.9 ± 5.8 | 71.4 ± 8.5 |
| Rat 5 | 12 | 55 | 11.60 ± 12.84 | 30.09 ± 9.83 | 39.18 ± 15.81 | 65.0 ± 7.1 | 74.6 ± 9.6 |
| **Mean** | **7.6** | **31.6** | **9.58 ± 9.19** | **27.72 ± 5.68** | **34.52 ± 9.89** | **62.2 ± 5.5** | **70.0 ± 8.2** |

For each animal we display the number of vascular surges and REM episodes and counted the mean and standard-deviation of five parameters including: duration of VS, mean and maximum surge intensity (in % CBV), mean and maximum active pixels ratio (in % of active pixels). The surge intensity is computed by averaging the CBV over all brain pixels for the whole VS duration. The activity ratio is computed by counting the number of active pixels (that is the number of pixels that cross the activity threshold as defined in Supplementary Fig. 2 and Methods). Although VS events show some intrinsic variability (in terms of duration for instance), these five parameters are very consistent across animals. We excluded two animals for which the total number of VS events was too low (lower than 5)

Dorsal Thalamus: from 37.91 to 47.81%; Cortex: from 17.24 to 37.47%) and that confidence intervals of CBV activation across sleep states were remarkably consistent across animals. We then assessed functional connectivity across states and found the same patterns for NREM and AW, consisting of a co-active cortex, thalamus and hippocampus, whereas functional coupling across regions was stronger in AW and strongest in REM (QW: Rpear_mean = .76, std = .11; NREM: Rpear_mean = .73, std = .07; AW: Rpear_mean = .88, std = .11; REM: Rpear_mean = .93, std = .04) [Fig. 2b]. Likewise, we computed confidence intervals for the analysis of the ratios of increases in functional coupling, for four animals across coronal planes. We found stronger inter-individual variability especially between AW-QW functional coupling, but all animals displayed stronger cortex-hippocampus, thalamus-cortex, and hippocampus-thalamic coupling during REMS than any other sleep and wake state, confirming the fact that functional coupling between distant brain areas increases during REMS [Supplementary Table 1]. Overall, AW/QW amplification resulted in a 6.0% increase, while REM/QW reached a 15.9% increase in the hippocampus; this level increased to 31.9% when considering phasic REM epochs [Fig. 2c and Supplementary Tables 2–3]. Not all regions show the same amplification ratio between the AW, REM and REM-phasic epochs, the strongest being observed in the dorsal hippocampus, dorsal thalamus, and retrosplenial cortex. These results show that REMS is a state of vascular hypersynchrony that is characterized by phasic massive spikes in CBV levels and is strongest in hippocampal regions.

**Band-specific Hippocampal LFP bursts precede vascular surges**. We found robust LFP precursors to VS in the form of a theta (6–10 Hz) power increase and acceleration accompanied by mid (50–100 Hz) and fast gamma bursts (100–150 Hz) [Fig. 3a, b]. Such phasic electrographic events have been described previously in rodents and cats, where they are thought to transiently synchronize hippocampal networks[18,19] and were observed preceding the vast majority of the VS events in all brain regions. Furthermore, we isolated the peak amplitude of each electrographic burst that we compared to its vascular counterpart. We found that vascular amplitude strongly correlated with LFP power in the mid (50–100 Hz) ($r = 0.771$, $t$-student = 20.5, $p < 10^{-3}$) and high (100–150 Hz) gamma ($r = 0.788$, $t$-student = 21.8, $p < 10^{-3}$) bands, but only moderately in the theta band ($r = 0.321$, $t$-student = 8.63, $p < 10^{-3}$) and not at all in the low-gamma (20–50 Hz) band ($r = 0.025$, $t$-student = 0.58, n.s.) [Fig. 3c]. We generalized this approach by thoroughly searching each band of the LFP between 1 and 250 Hz, yielding similar results and showing a prominent peak in the 80–110 Hz band [Fig. 3d and Supplementary Fig. 5]. The analysis of the delays between LFP gamma

bursts and CBV events relative to the peak of theta, revealed that gamma power was stronger in the second half of each theta burst (low-gamma: $\Delta t1 = 300$ ms; mid-gamma: $\Delta t2 = 340$ ms; high-gamma: $\Delta t3 = 280$ ms), and that thalamic regions ($\Delta t4 = 1303$ ms, sem = 3.35 ms, $Z$-score = 21.1, $p < 10^{-3}$) peaked before hippocampal ($\Delta t5 = 1458$ ms, sem = 3.37 ms, $Z$-score = 22.7, $p < 10^{-3}$) and cortical regions ($\Delta t6 = 1493$ ms, sem = 3.60 s, $Z$-score = 22.0, $p < 10^{-3}$) [Supplementary Table 4], revealing a sequential activation associated with gamma bursts [Fig. 3e and Supplementary Fig. 6]. Altogether, these results finely characterize the spatiotemporal hemodynamics associated with fast gamma oscillations during phasic REMS.

We investigated the co-occurrence of LFP oscillations and vascular activation in deeper details, first to confirm which LFP band best described the fluctuations observed in the regional CBV signal but also to investigate the frequency of occurrence of the LFP burst activity preceding each VS event [Fig. 4]. We used two complementary approaches, first by detecting VS in each of the three main regions (thalamus, hippocampus, cortex) and searching for LFP events in the timing window [−4.0 s; + 1.0 s] preceding the CBV peak. We found strongest co-occurrences (over 75% in the mid and fast gamma bands for all three regions) with highest coupling in the hippocampus (Low-Gamma: 368/611; Mid-Gamma: 441/611; High-Gamma: 458/611; Theta: 428/611). The second approach consisted in detecting LFP burst events and searching for CBV peaks in the timing window following electrical activity. This yielded similar results, albeit with weaker co-occurrence ratios (between 65 and 70% for all three main regions) showing a weak tendency for LFP activity to occur more often preceding VS events than for CBV activity to follow electrical activation.

**Fast gamma events lead vascular activity by 1.5 to 2 s**. In order to relate our data to previous work showing that fluctuations in gamma power trigger an increase in arteriole diameter in 2 s and changes in oxygenation in 3 s in awake head-fixed mice[29], we performed cross-correlations analysis between the LFP envelopes and the regional CBV averages in all three main regions [Fig. 5]. Importantly, this analysis is somewhat different from the LFP-CBV peak co-occurrence analysis performed previously as it does not consider individual neural and vascular events, but computes a cross-correlation function for each REM episode individually thus providing a more robust measure of neurovascular interactions, by smoothing out the effects of individual events. We computed the cross-correlations between four LFP bands (Low-Gamma, Mid-Gamma, High-Gamma, Theta) and three regional CBV signals (Cortex, Thalamus, Hippocampus) for 28 recording episodes over seven animals and displayed individual cross-correlation functions for all resulting pairs [Fig. 5a]. We found

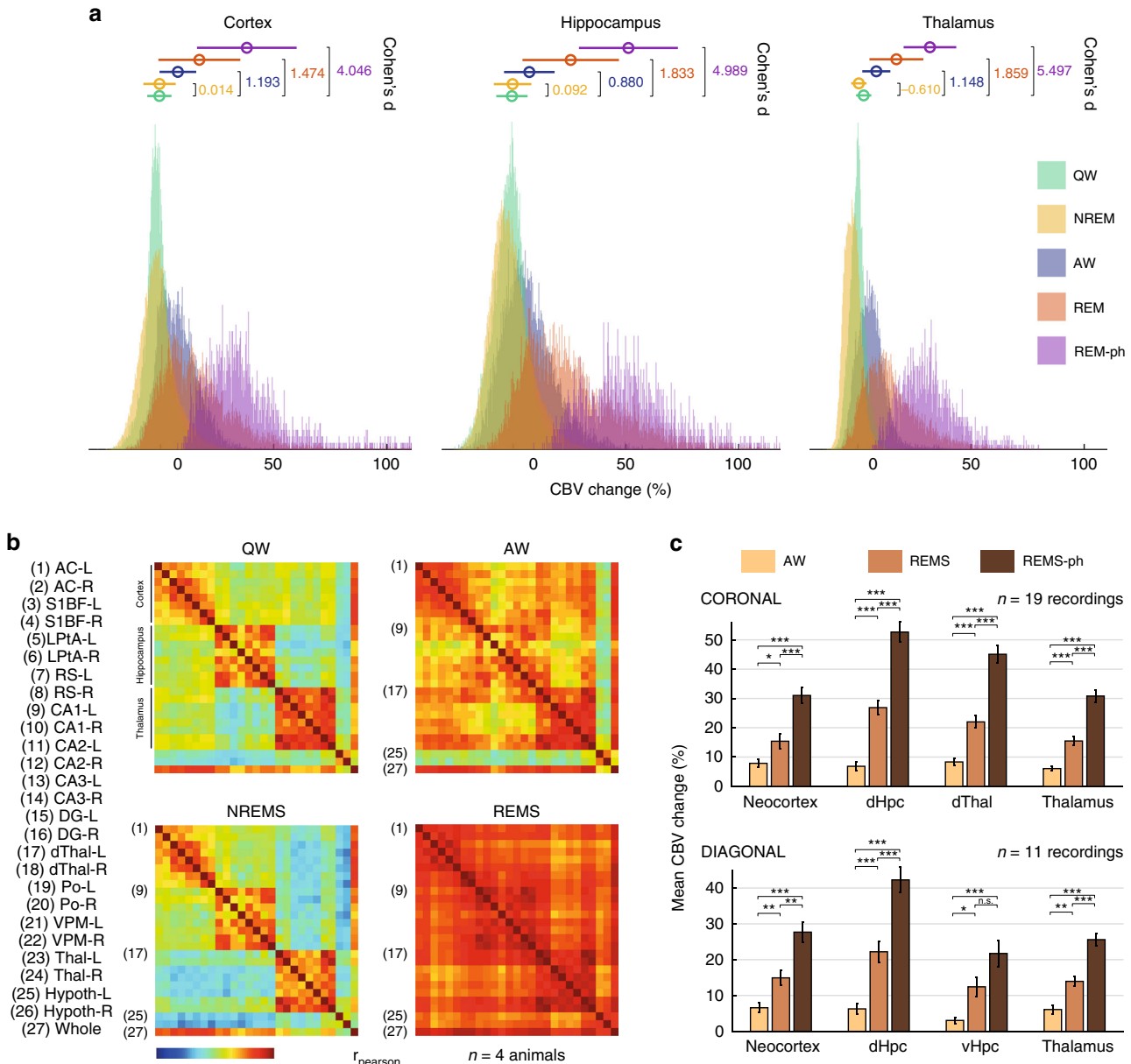

**Fig. 2** REMS hyperemia is generalized but strongest in the hippocampus. **a** Vascular distributions for all animals in three regions (cortex, hippocampus, thalamus) for five different states: quiet wake (QW), active wake (AW), non-REM sleep (NREMS), REM sleep (REMS), and phasic REM sleep (REMS-ph). QW and NREM overlap except in the thalamus. AW is an intermediate state between QW and REM. Hippocampus shows the clearest separation between AW, REMS, and REM-phasic distribution. Horizontal axis: CBV amplitude (% from baseline). Vertical axis: Event count. Circles and horizontal bars show the mean and standard-deviation for each behavioral state. Cohen's d relative to QW are given for all three regions. **b** Functional connectivity matrices for all four sleep stages, averaged over all recordings (n = 4 animals). Each matrix displays the Pearson correlation between all pairs of regional CBV variables. Note the similarity between QW and NREM consisting of three co-active blocks including cortical, hippocampal, and thalamic regions and the gradual synchronization of all regional signals during AW and REMS. **c** Vascular amplification obtained by averaging AW, REMS, and REMS-ph mean values (Cortex, Hippocampus, Thalamus, and Whole) for all recordings, grouped by recording plane. Strikingly, all four regions consistently display vascular amplification, with highest ratio obtained by comparing phasic REMS (REMS-ph) and quiet wake (QW) in the dorsal hippocampus

peaks in the positive LFP-CBV delay timing window moderately for the theta band and very robustly for the mid and high-gamma bands with LFP leading CBV response by a delay in a time interval varying between 1 to 3 s. Low-Gamma showed highly variable cross-correlation profile. We extracted a mean cross-correlation profile to compute an average LFP-CBV delay for all 12 pairs [Fig. 5b]. As expected this showed a strong neurovascular coupling in the mid and fast gamma bands, with mean cross-correlations maxima between 0.6 and 0.7 for all three main

regions (Thalamus: Rmean = 0.64 obtained at lag = 1.35 s; Hippocampus: Rmean = 0.69 obtained at lag = 1.52 s; Hippocampus: Rmean = 0.64 obtained at lag = 1.65 s relatively to high-gamma oscillations). Finally, we display the histogram count of the LFP-CBV delays for all three main regions and cortex relatively to high-gamma oscillations [Fig. 5c]. This confirmed that neurovascular coupling is faster in the thalamic regions and slower in the cortical regions compared to the hippocampus. Although the mean delays (around 1.5 s) we found were somewhat shorter than

**Table 2 Inter-individual analysis of CBV distributions across brain regions and sleep states**

| CORTEX | QW | | | | NREM | | | | AW | | | | REM | | | | REM-TONIC | | | | REM-PHASIC | | | |
|---|---|---|---|---|---|---|---|---|---|---|---|---|---|---|---|---|---|---|---|---|---|---|---|---|
| Animal | Mean | IC_95% | | p | Mean | IC_95% | | p | Mean | IC_95% | | p | Mean | IC_95% | | p | Mean | IC_95% | | p | Mean | IC_95% | | p |
| Rat1 (7) | 0.1 | −0.3 | 0.4 | - | 1.6 | 1.2 | 2.0 | 0.00 | 4.5 | 3.2 | 5.7 | 0.00 | 13.3 | 10.8 | 15.9 | 0.00 | 7.5 | 5.9 | 9.2 | 0.00 | 26.8 | 22.7 | 30.9 | 0.00 |
| Rat2 (7) | −0.2 | −0.8 | 0.4 | - | −2.7 | −3.0 | −2.4 | 0.00 | 3.3 | 1.6 | 5.0 | 0.00 | 10.0 | 8.2 | 11.9 | 0.00 | 4.7 | 3.0 | 6.3 | 0.00 | 17.2 | 14.3 | 20.2 | 0.00 |
| Rat3 (9) | 0.6 | 0.1 | 1.0 | - | 0.4 | 0.1 | 0.8 | 0.61 | 11.1 | 9.6 | 12.6 | 0.00 | 20.8 | 17.3 | 24.3 | 0.00 | 11.2 | 9.4 | 12.9 | 0.00 | 37.5 | 31.7 | 43.2 | 0.00 |
| Rat4 (5) | −1.0 | −1.8 | −0.1 | - | −3.2 | −3.5 | −2.8 | 0.00 | 8.2 | 6.7 | 9.7 | 0.00 | 12.6 | 9.9 | 15.2 | 0.00 | 9.2 | 6.9 | 11.6 | 0.00 | 27.3 | 22.2 | 32.4 | 0.00 |
| Rat5 (3) | 3.0 | 1.4 | 4.7 | - | 1.2 | 0.5 | 1.8 | 0.01 | 11.3 | 9.1 | 13.5 | 0.00 | 15.6 | 13.2 | 17.9 | 0.00 | 14.1 | 11.8 | 16.4 | 0.00 | 30.8 | 24.2 | 37.3 | 0.00 |

| dHPC | QW | | | | NREM | | | | AW | | | | REM | | | | REM-TONIC | | | | REM-PHASIC | | | |
|---|---|---|---|---|---|---|---|---|---|---|---|---|---|---|---|---|---|---|---|---|---|---|---|---|
| Animal | Mean | IC_95% | | p | Mean | IC_95% | | p | Mean | IC_95% | | p | Mean | IC_95% | | p | Mean | IC_95% | | p | Mean | IC_95% | | p |
| Rat1 (7) | 0.4 | −0.1 | 1.0 | - | −0.2 | −0.6 | 0.1 | 0.09 | 3.7 | 2.2 | 5.3 | 0.00 | 20.1 | 16.6 | 23.7 | 0.00 | 11.6 | 8.9 | 14.2 | 0.00 | 40.0 | 35.6 | 44.4 | 0.00 |
| Rat2 (7) | −0.8 | −1.4 | −0.2 | - | −4.1 | −4.5 | −3.7 | 0.00 | 5.3 | 3.1 | 7.5 | 0.00 | 19.5 | 15.0 | 24.0 | 0.00 | 10.3 | 7.3 | 13.4 | 0.00 | 44.1 | 38.3 | 50.0 | 0.00 |
| Rat3 (9) | −0.4 | −1.1 | 0.2 | - | 2.9 | 2.3 | 3.5 | 0.00 | 8.6 | 6.8 | 10.3 | 0.00 | 28.9 | 25.3 | 32.5 | 0.00 | 18.9 | 16.2 | 21.6 | 0.00 | 47.4 | 41.3 | 53.4 | 0.00 |
| Rat4 (5) | −0.7 | −1.6 | 0.2 | - | 1.1 | 0.6 | 1.5 | 0.00 | 6.6 | 5.5 | 7.6 | 0.00 | 29.2 | 25.1 | 33.3 | 0.00 | 23.1 | 19.7 | 26.5 | 0.00 | 55.4 | 48.2 | 62.5 | 0.00 |
| Rat5 (3) | 3.9 | 1.9 | 5.8 | - | 2.8 | 1.9 | 3.7 | 0.08 | 11.6 | 8.8 | 14.4 | 0.00 | 22.6 | 19.0 | 26.1 | 0.00 | 20.5 | 17.0 | 24.0 | 0.00 | 49.8 | 33.4 | 66.2 | 0.00 |

| dTHAL | QW | | | | NREM | | | | AW | | | | REM | | | | REM-TONIC | | | | REM-PHASIC | | | |
|---|---|---|---|---|---|---|---|---|---|---|---|---|---|---|---|---|---|---|---|---|---|---|---|---|
| Animal | Mean | IC_95% | | p | Mean | IC_95% | | p | Mean | IC_95% | | p | Mean | IC_95% | | p | Mean | IC_95% | | p | Mean | IC_95% | | p |
| Rat1 (7) | 0.1 | −0.2 | 0.5 | - | −0.5 | −0.9 | −0.2 | 0.26 | 7.4 | 6.3 | 8.4 | 0.00 | 21.1 | 17.9 | 24.3 | 0.00 | 13.4 | 10.6 | 16.2 | 0.00 | 37.9 | 33.9 | 41.9 | 0.00 |
| Rat2 (7) | −0.3 | −0.8 | 0.1 | - | −1.0 | −1.3 | −0.7 | 0.03 | 9.7 | 7.0 | 12.5 | 0.00 | 23.0 | 19.2 | 26.7 | 0.00 | 14.1 | 11.4 | 16.8 | 0.00 | 39.4 | 36.5 | 42.3 | 0.00 |
| Rat3 (9) | −0.9 | −2.0 | 0.1 | - | −2.0 | −2.6 | −1.4 | 0.03 | 6.0 | 3.1 | 8.8 | 0.00 | 24.1 | 19.8 | 28.4 | 0.00 | 14.1 | 12.2 | 16.1 | 0.00 | 47.8 | 39.2 | 56.4 | 0.00 |
| Rat4 (5) | −0.5 | −1.2 | 0.2 | - | −3.4 | −3.8 | −3.1 | 0.00 | 8.3 | 7.4 | 9.3 | 0.00 | 17.2 | 12.5 | 21.9 | 0.00 | 11.8 | 7.9 | 15.7 | 0.00 | 40.0 | 35.6 | 44.4 | 0.00 |
| Rat5 (3) | 2.7 | 1.0 | 4.5 | - | −2.8 | −3.7 | −1.9 | 0.00 | 10.3 | 7.1 | 13.5 | 0.00 | 18.3 | 14.9 | 21.7 | 0.00 | 16.5 | 12.9 | 20.1 | 0.00 | 35.0 | 20.8 | 49.2 | 0.00 |

| THAL | QW | | | | NREM | | | | AW | | | | REM | | | | REM-TONIC | | | | REM-PHASIC | | | |
|---|---|---|---|---|---|---|---|---|---|---|---|---|---|---|---|---|---|---|---|---|---|---|---|---|
| Animal | Mean | IC_95% | | p | Mean | IC_95% | | p | Mean | IC_95% | | p | Mean | IC_95% | | p | Mean | IC_95% | | p | Mean | IC_95% | | p |
| Rat1 (7) | 0.1 | −0.1 | 0.4 | - | −0.5 | −0.7 | −0.3 | 0.01 | 5.6 | 4.8 | 6.5 | 0.00 | 16.0 | 13.9 | 18.2 | 0.00 | 11.2 | 9.3 | 13.0 | 0.00 | 27.4 | 24.4 | 30.3 | 0.00 |
| Rat2 (7) | −0.1 | −0.5 | 0.2 | - | −2.0 | −2.3 | −1.7 | 0.00 | 6.0 | 4.9 | 7.2 | 0.00 | 14.2 | 11.6 | 16.9 | 0.00 | 10.2 | 7.4 | 13.0 | 0.00 | 25.4 | 22.7 | 28.0 | 0.00 |
| Rat3 (9) | 0.4 | 0.0 | 0.8 | - | −0.6 | −0.9 | −0.3 | 0.00 | 11.8 | 9.5 | 14.0 | 0.00 | 17.4 | 15.3 | 19.6 | 0.00 | 11.5 | 9.9 | 13.2 | 0.00 | 28.7 | 24.6 | 32.7 | 0.00 |
| Rat4 (5) | −0.6 | −1.1 | 0.0 | - | −3.1 | −3.2 | −2.9 | 0.00 | 4.0 | 3.4 | 4.7 | 0.00 | 12.8 | 10.8 | 14.9 | 0.00 | 9.7 | 7.9 | 11.6 | 0.00 | 26.6 | 23.7 | 29.6 | 0.00 |
| Rat5 (3) | 1.5 | 0.5 | 2.4 | - | −2.3 | −2.6 | −1.9 | 0.00 | 6.1 | 4.5 | 7.7 | 0.00 | 12.7 | 11.0 | 14.5 | 0.00 | 11.9 | 10.1 | 13.7 | 0.00 | 24.3 | 17.5 | 31.1 | 0.00 |

| WHOLE | QW | | | | NREM | | | | AW | | | | REM | | | | REM-TONIC | | | | REM-PHASIC | | | |
|---|---|---|---|---|---|---|---|---|---|---|---|---|---|---|---|---|---|---|---|---|---|---|---|---|
| Animal | Mean | IC_95% | | p | Mean | IC_95% | | p | Mean | IC_95% | | p | Mean | IC_95% | | p | Mean | IC_95% | | p | Mean | IC_95% | | p |
| Rat1 (7) | 0.1 | −0.2 | 0.3 | - | 1.0 | 0.7 | 1.2 | 0.00 | 4.7 | 3.6 | 5.9 | 0.00 | 14.4 | 12.2 | 16.6 | 0.00 | 9.0 | 7.5 | 10.5 | 0.00 | 26.8 | 24.5 | 29.1 | 0.00 |
| Rat2 (7) | −0.3 | −0.7 | 0.2 | - | −2.5 | −2.7 | −2.2 | 0.00 | 3.2 | 1.7 | 4.6 | 0.00 | 11.6 | 10.2 | 12.9 | 0.00 | 7.1 | 5.8 | 8.4 | 0.00 | 17.7 | 15.8 | 19.7 | 0.00 |
| Rat3 (9) | 0.5 | 0.1 | 0.9 | - | 0.7 | 0.4 | 1.0 | 0.17 | 9.4 | 8.0 | 10.8 | 0.00 | 21.1 | 18.2 | 24.1 | 0.00 | 12.7 | 10.9 | 14.5 | 0.00 | 36.0 | 31.4 | 40.7 | 0.00 |
| Rat4 (5) | −0.7 | −1.3 | 0.0 | - | −1.5 | −1.7 | −1.2 | 0.00 | 5.4 | 4.5 | 6.2 | 0.00 | 15.9 | 13.5 | 18.4 | 0.00 | 12.5 | 10.4 | 14.6 | 0.00 | 31.2 | 26.8 | 35.6 | 0.00 |
| Rat5 (3) | 2.2 | 1.0 | 3.4 | - | 0.2 | −0.3 | 0.8 | 0.00 | 8.5 | 6.8 | 10.1 | 0.00 | 14.5 | 12.5 | 16.5 | 0.00 | 13.3 | 11.4 | 15.3 | 0.00 | 28.4 | 22.2 | 34.5 | 0.00 |

The analysis is performed for five brain regions: cortex, dorsal Hippocampus (dHpc), dorsal Thalamus (dThal), Whole Thalamus (Thal), and Whole-brain (Whole). For each brain region, one row represents a single animal (the corresponding number of recordings is given between brackets) and a set of four columns corresponds to a single sleep state. For each sleep state, the left (bold) number is the mean CBV level, the second and third figures are the 95% confidence interval and the last value is the corresponding p-value for the observed sleep state to be drawn from the QW distribution. CBV levels are expressed in % of change relative to a baseline (3 first minutes of QW)

the ones found in the above study (about 1.9–2 s)[29], we found significant inter-episodes variability, which might explain this discrepancy, apart from the notable experimental set-up differences (species, head-fixed vs. head-free) and arousal state (sleep vs. wakefulness). Importantly, we monitored the increases in fractional cerebral blood volume occurring in arterioles but excluding parenchyma capillaries where neurovascular interactions can be triggered first[30]. This study confirms that vascular activations in medium to large vessels trail fast gamma oscillations during spontaneous sleep by similar delays as what was observed previously.

## Discussion

Our findings bring new information about vascular activity levels and dynamics across sleep stages and during REMS episodes in particular. Several models of REMS function have been proposed including the replay of extended place cell sequences[31], improving creativity[32], or unlearning irrelevant memories[33]. Our findings of high levels of vascular activity, tightly coupled to fast gamma oscillations and distributed on a large scale may help refine these models. We observed massive brain-wide amplification during REMS compared to active states such as running. To the best of our knowledge, such activations have surprisingly not yet been detected nor reported over the past 50 years, other than using autoradiography[21], a technique requiring sacrifice and later criticized by one of its originators[34]. Several reasons may explain this: first, physiological whole-brain data, especially in deep structures, is difficult to obtain by direct techniques without anesthesia. Secondly, the lack of sensitivity in fMRI/PET studies, the difficulty of reliably performing EEG-fMRI

paradigms, device noise and intrinsic caveats associated with sleep deprivation protocols represent a bottleneck for such animal sleep studies. Thirdly, sleep research has focused on NREMS due to the importance of sharp-wave/ripples and place cell coding (which gives a physical correlate of neuronal activity), leaving place cell recording during REMS scarce[31]. Last, the rarity of REMS episodes in rodents and the fact that REMS episodes are more frequent and longer at the end of the night in human sleep is a challenge in multimodal imaging.

We have also demonstrated a robust association between fast gamma oscillations and whole-brain vascular hyperactivity, which was also observed under anesthesia in previous EEG-fMRI and intrinsic optical signal (IOS)-fMRI studies[35,36]. The fast gamma oscillations we observed are significantly faster than those observed in cats, monkeys, or humans, but have previously been described in rodents[37]. We investigated the coupling pattern between theta phase and gamma power by computing the mean power of gamma envelope relative to theta phase and found that gamma power was strongly modulated by theta phase both during REMs and AW [Supplementary Fig. 1]. During REMS, mid and fast gamma peaked earlier (in the [90°–180°] range relative to theta trough, or ascending theta phase) than slow gamma oscillations (in the [180°–270°] range relative to theta trough, or descending theta phase). Based on this observation, we assume the mid-gamma oscillations we observed to be triggered by direct entorhinal input to the CA1 region, as opposed to slow gamma (30–50 Hz) putatively triggered by CA3 input[18,37,38]. This opens the door to the pharmacological inhibition and optogenetic manipulation of such oscillations for investigations into the global effects of neurovascular coupling. Moreover, this challenges the

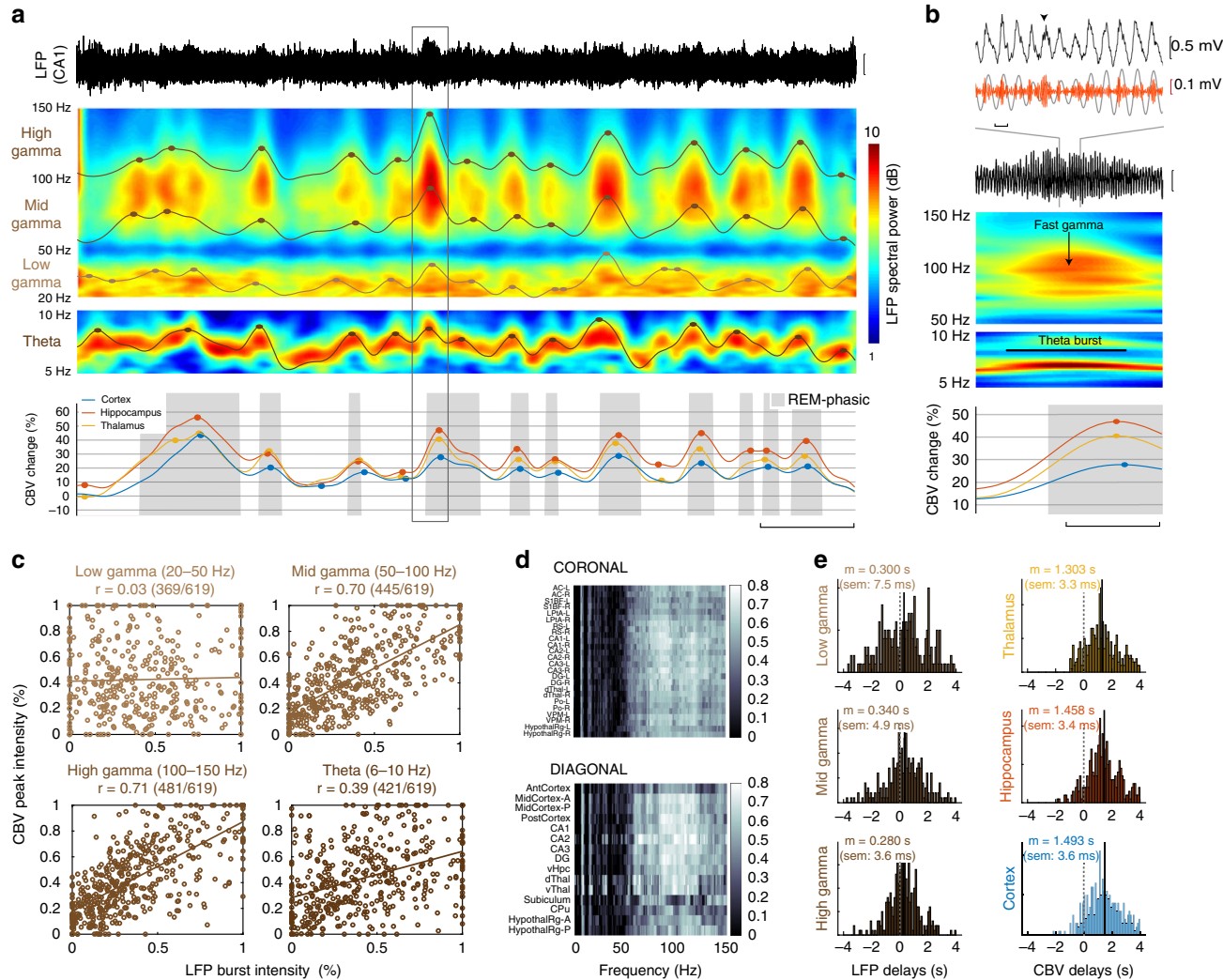

**Fig. 3** Hippocampal theta and fast gamma bursts precede vascular surges. **a** CA1 Hippocampal LFP trace (top) time-frequency spectrograms with four LFP envelopes (middle) and regional CBV dynamics (bottom) for a typical REM sleep episode. Solid circles mark the regional peaks for LFP and CBV signals. Note that each vascular surge (gray box) is preceded by sustained activity in the theta, mid-gamma, and high-gamma bands. Note that we can pair up LFP peaks (brown/black dots) with corresponding vascular peaks (color dots) to investigate LFP-CBV coupling. Scale bar: 30 s. **b** Typical LFP precursor to a vascular surge (extended box shown in **a**). At the onset of the VS, CA1 hippocampal LFP shows prominent theta and fast gamma bursts, occurring on the ascending phase/peak of the theta cycle (gray curve). Upper scale bar: 100 ms. Lower scale bar: 5 s. **c** Scatter plot showing the correspondence between LFP peaks (x-axis) corresponding CBV peak (y-axis), for the four frequency bands shown in **a**. Note the strong correlations in the mid (50–100 Hz) and high (100–150 Hz) gamma bands, showing that hippocampal gamma predicts the amplitude of vascular amplification. The numbers in bracket specify the pairing ratio between LFP and CBV peaks. **d** Exhaustive analysis of LFP-CBV correlations for all brain regions and LFP frequency bands. Horizontal axis: LFP bands, Vertical axis: regional CBV variables. Each coefficient of matrices is obtained by generalizing the approach in **c**. Note that highest correlations are obtained for gamma-mid band (50–100 Hz) in the hippocampal (DG-CA1-CA3) and dorsal thalamic (dThal) regions. **e** Histograms of delays for the three LFP gamma sub-bands and the three CBV regions, relative to theta peaks. Note that cortical delays ($m = 1.493$ s) are longer than thalamic ($m = 1.303$ s) and hippocampal ($m = 1.458$ s) delays

notion that fast oscillations are a proxy for local information processing, suggesting instead that they trigger activation over largely distributed neural networks, possibly through neurovascular interactions involving calcium spikes.

Although we did not question their function, we hint that the spatiotemporal dynamics of these vascular surges are somewhat related with the neural replay of previous experience, in our case, a running task on a 2.4-m linear track. Firstly, theta and gamma bursts observed during phasic REM periods are the same duration and amplitude as the theta and gamma bursts observed when these animals were actually running, which is consistent with previous observations by Montgomery and colleagues[18]. Secondly, the vascular patterns observed during REMS involved the

same highly responsive regions in a similar dynamic sequence (sequential activation of dorsal thalamus, dorsal hippocampus, and cortex) as observed and described during active running on a similar track[27]. Thirdly, based on recent studies identifying high-frequency oscillations as the neural correlates of dreaming in humans[39], it is possible that fast gamma oscillations and subsequent vascular surges, functionally coupled during REMS, are involved in the generation of dream content in animals. Taken together, these elements suggest a potential link in our experimental configuration between REMS, VS, and the replay of past experience. The combined fUS-EEG method is a unique tool to answer this question and investigate the brain dynamics of other complex behaviors such as conditioning, learning or dreaming.

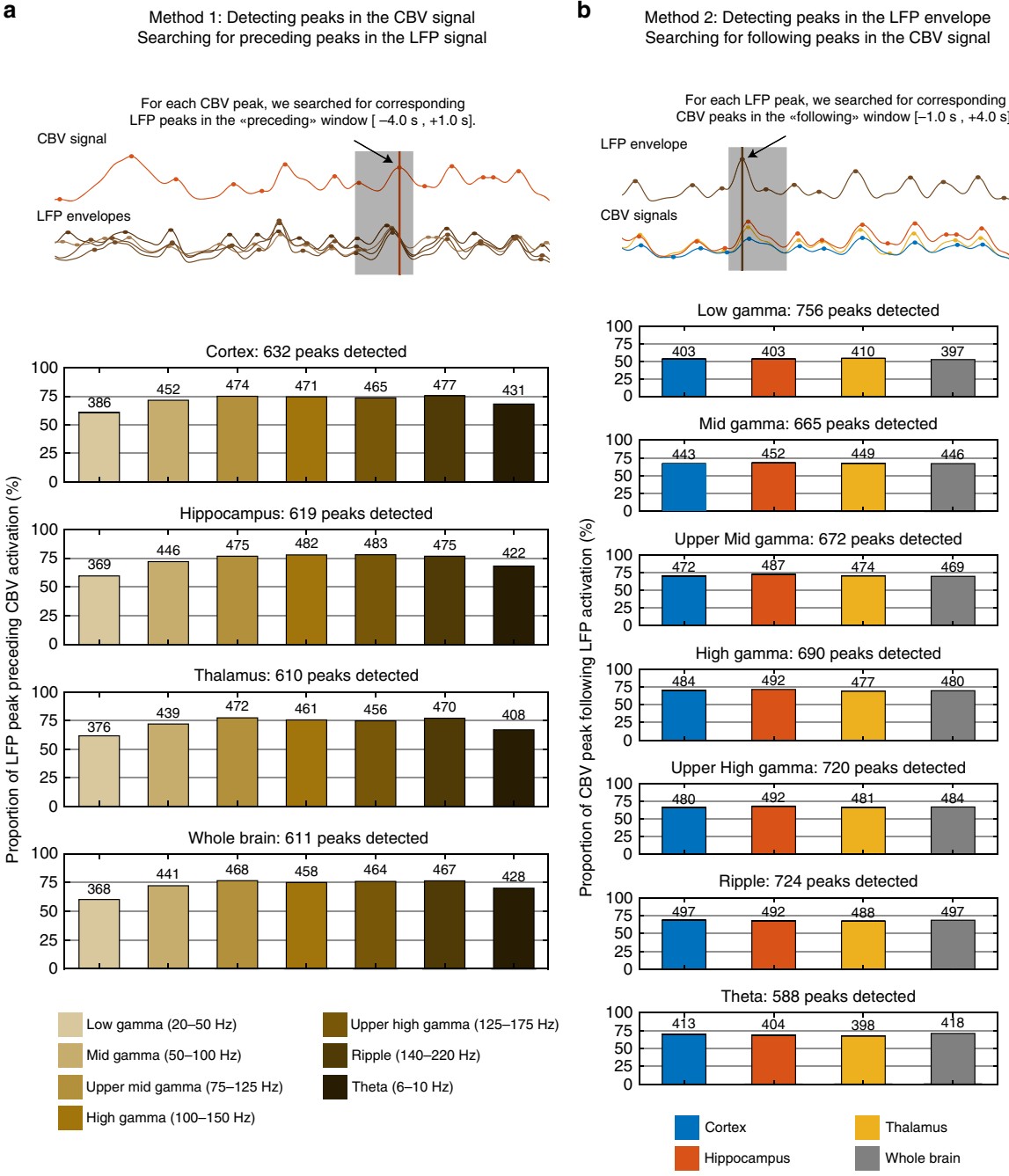

**Fig. 4** Co-occurrence analysis of LFP and CBV activation peaks. **a** We first detected peaks for each of the four regional CBV averages (cortex, hippocampus, thalamus, and whole). For each detected peak, we searched for peaks in the seven LFP envelope signals (Low-Gamma, Mid-Gamma, Upper Mid-Gamma, High-Gamma, Upper High-Gamma, Ripple, and Theta) in the timing window [−4.0 s, +1.0 s] preceding the CBV peak. If more than one peak was found, the closest was selected. The bar graphs display the proportion of LFP peaks found for each LFP-CBV signal pair (absolute number are given on top of each bar). For all regions, the highest LFP-CBV co-occurrence ratio is found in the high-gamma band. **b** We performed the complementary approach by detecting LFP peaks and searching the timing window [−1.0 s, +4.0 s] following LFP peak. The highest LFP-CBV co-occurrence ratio is found in the hippocampus no matter the LFP band. Analysis performed for $n = 7$ animals overs 28 recordings. The corresponding LFP-CBV correlations and delay histograms are shown in the following Supplementary Figs. 5 and 6

Extrapolation of our results to human physiology should be done cautiously. Human sleep structure differs strongly from that of rodents. It is unclear whether such high-amplitude vascular events exist in humans, although BOLD-based connectivity patterns spread brain-wide during REMS[24] and newborn sleep display high-amplitude vascular activity during early active sleep[40]. At the experimental level, the current difficulties (absorption by the bone, lack of sensitivity) to transfer functional ultrasound imaging to healthy adult human subjects is an obstacle to investigate brain-wide human sleep dynamics and will require alternative strategies such as the use of contrast agents or aberration correction[41]. Our current design also suffers some limitations as it relies on power Doppler to estimate relative cerebral blood volume. In our current set-up, and contrary to earlier implementations of fUS that measured the fraction of CBV flowing with an axial velocity higher than 4 mms$^{-1}$ (refs. [26,42]),

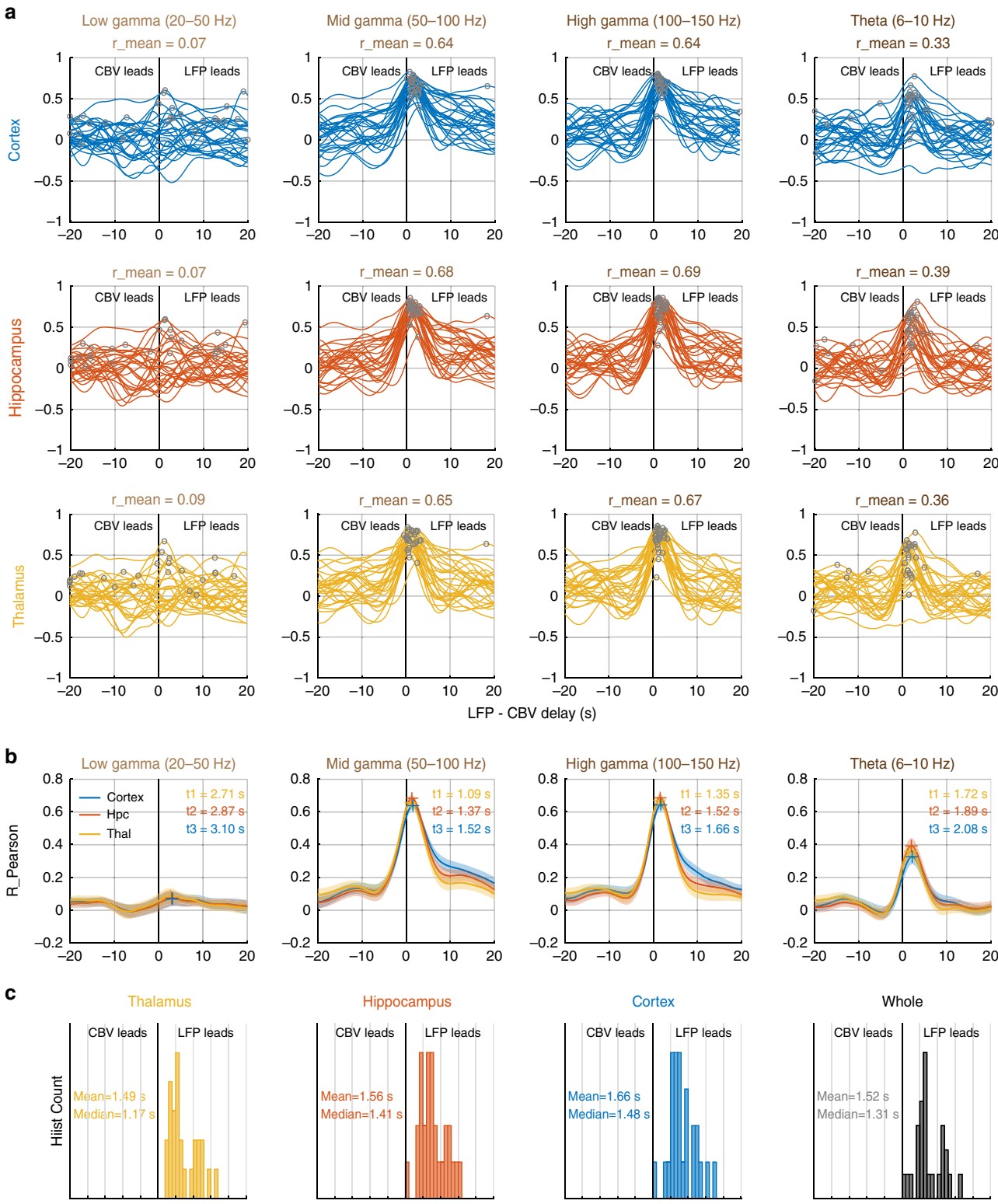

**Fig. 5** Fast gamma events lead hemodynamic patterns by 1.5 to 2 s. **a** Cross-correlation functions between four LFP envelope signals: Theta (6–10 Hz), Low-Gamma (20–50 Hz), Mid-Gamma (50–100 Hz), and High-Gamma (100–150 Hz) and three regional CBV signals (Thalamus, Hippocampus, Cortex) for $n=28$ REM episodes (six rats) lasting >30 s. Note that low-gamma shows no coupling with CBV signals in either region, theta shows moderate coupling, whereas both mid and high-gamma bands show strong robust coupling and reproducible LFP-CBV delays in all three regions. Each line shows one recording; gray dots show the peak of the cross-correlation function, which we used to extract the LFP-CBV delay of neurovascular coupling. **b** Mean cross-correlation functions for the 28 recordings shown above. Note that the delays follow the sequential activation pattern Thalamus-Hippocampus-Cortex described in Fig. 3. Color crosses display the peak of the mean cross-correlation function. **c** Histogram count of LFP-CBV delays extracted from the peak of cross-correlation function of high-gamma events. Note that LFP leads CBV for all recordings and shows some variability in the 1–3 s lag window. None of these delays was negative showing that LFP leads vascular activations in all 28 recordings

ultrafast Doppler combined with the use of SVD spatiotemporal filter enables to reach a detection limit down to 1 mms$^{-1}$[43]. This corresponds to speed in venules and arterioles but may exclude capillaries where numerous neurovascular processes occur[44]. Moreover, power Doppler is essentially blind to blood vessels whose direction is quasi parallel to the transducer array (+/−15°), which thus do not contribute to the measured CBV.

Ultimately, the amplitude and spatial extent of the massive hemodynamic patterns unveiled by fUS imaging both confirm that REMS is a very intense brain state— with an apparently higher homeostatic point than wake—but also fundamentally questions the putative function of REMS. These vascular patterns suggest that the underlying mechanisms are energetically costly in terms of neurovascular regulation and thus must have been preserved only due to a significant evolutionary benefit, yet to be determined.

## Methods

**Animal surgery.** All animals received humane care in compliance with the European Communities Council Directive of 2010 (2010/63/EU), and the institutional and regional committees for animal care approved the study. Adult Sprague Dawley rats aged 10–12 weeks underwent surgical craniotomy and implant of an ultrasound-clear prosthesis. Anesthesia was induced with 2% isoflurane and maintained with ketamine/xylazine (80/10 mg kg$^{-1}$), while body temperature was maintained at 36.5 °C with a heating blanket (Bioseb, France). A sagittal skin incision was performed across the posterior part of the head to expose the skull. We excised the parietal and frontal flaps by drilling and gently moving the bone away from the dura mater. The opening exposed the brain between the olfactory bulb and the cerebellum, from bregma + 6.0 to bregma −8.0 mm, with a maximal width of 14 mm. Electrodes were implanted stereotaxically and anchored on the edge of the flap. A prosthetic skull was sealed in place with acrylic resin (GC Unifast TRAD), and the residual space was filled with saline. We chose a prosthesis approach that offers a larger field of view and prolonged imaging condition over 4–6 weeks compared to the thinned bone approach[45]. The prosthetic skull is composed of polymethylpentene (Goodfellow, Huntington UK, goodfellow.com), a standard biopolymer used for implants. This material has tissue-like acoustic impedance that allows undistorted propagation of ultrasound waves at the acoustic gel-prosthesis and prosthesis-saline interfaces. The prosthesis was cut out of a film of 250 μm thickness and permanently sealed to the skull. Particular care was taken not to tear the dura to prevent cerebral damage. The surgical procedure, including electrode implantation, typically took 4–6 h. Animals recovered quickly and were used for data acquisition after a conservative 1-week resting period.

**Electrode design and implantation.** Electrodes are based on linear polytrodes made of bundles of insulated tungsten wires. The difference with a standard design is a 90°-angle elbow that is formed prior to insertion in the brain[27]. This shape enabled anchoring of the electrodes on the skull anterior or posterior to the flap. Electrodes were implanted with stereotaxic positioning micromotion and anchored one after another. The prosthesis was then applied to seal the skull. Two epidural screws placed above the cerebellum were used as a reference and ground. Intra-hippocampal handmade electrode bundles were composed of 25 to 50 μm insulated tungsten wire soldered to miniature connectors. Four to six conductive ends were spaced 1 mm apart and glued to form 3-mm-long, 100–150-μm-diameter bundles. The bundles were lowered in the dorsal hippocampi at stereotaxic coordinates AP = −4.0 mm, ML = +/− 2.5 mm and DV = −1.5 mm to −4.5 mm relative to the Bregma.

**LFP acquisition.** LFP signals were collected from video-EEG device for offline processing. Intracranial electrode signals were fed through a high input impedance, DC-cut at 1 Hz, gain of 1000, 16-channel amplifier, and digitized at 20 kHz (Xcell, Dipsi, Cancale, France), together with a synchronization signal from the ultrasound scanner. Custom-made software based on LabVIEW (National Instruments, Austin, TX, USA) simultaneously acquired video from a camera pointed at the recording stage. A regular amplifier was used, and no additional electronic circuit for artifact suppression was necessary. A large bandwidth amplifier was used, which can record local field potentials in all physiological bands (LFP, 0.1–2 kHz). The spatial resolution of LFPs ranges from 250 μm to a few mm radius.

**Ultrasound acquisition.** Vascular images were obtained via the ultrafast compound Doppler imaging technique[26]. The probe was driven by a fully programmable GPU-based ultrafast ultrasound scanner (Aixplorer, Supersonic Imagine, Aix-en-Provence, France) relying on 24-Gb RAM memory. We acquired 400 ultrasound images at a 1 kHz frame rate for 200 ms, repeating every 1.0 s to 3.0 s. Each frame is a compound plane-wave frame, that is, a coherent summation of beamformed complex in phase/quadrature images obtained from the insonification

of the medium with a set of successive plane waves with specific tilting angles[46]. This compound plane-wave imaging technique enabled the recreation of a dynamic transmit focusing at all depths *a posteriori* in the entire field of view with few ultrasound emissions. Given the tradeoff between frame rate, resolution and imaging speed, a plane-wave compounding using five 5°-apart angles of insonification (from −10° to + 10°) was chosen. As a result, the pulse repetition frequency of the plane-wave transmissions was equal to 500 Hz. To discriminate blood signals from tissue clutter, the ultrafast compound Doppler frame stack was filtered, removing the $N = 60$ first components of the singular value decomposition, which optimally exploited the spatiotemporal dynamics of the full Doppler film for clutter rejection, largely outperforming conventional clutter-rejection filters used in Doppler ultrasound[47].

**Recording sessions.** Recording sessions were performed during continuous periods of 40–60 min that followed a basic track-running task. The task consisted of running along a 2.25-m-long linear track for water reward on both ends. The rats were placed under a controlled water restriction protocol (weight maintained between 85 and 90% of the normal weight) and trained before surgery to run back and forth on a linear track for water reinforcement. The track (225 × 20 cm) had 5-cm-high lateral walls and was placed 50 cm above the ground. Drops of water were delivered through two small tubes coming from the two end walls of the track. Each time the animal crossed the middle of the track, a single drop of water was delivered in alternate water tubes by opening an electronically controlled pair of solenoid valves. Daily training lasted 30 min. At the start of each recording session, to attach the ultrasound probe and to attach the EEG, the rats underwent brief anesthesia for 20–25 min with 2% isoflurane. Acoustic gel was applied on the skull prosthesis, and the probe was inserted into the probe holder. The gel did not dry out even for extended recordings of up to 6–8 h. The animals were allowed to recover for 40 min before starting the recording session. A typical session included a 30–40 min running period and 1-h sleep for a duration of approximately 3 h. The post-task sleep sessions were initiated 10 min after the end of the running task. In total, we recorded from seven animals over the coronal and diagonal planes for a total of 35 running-sleep sessions; thus, 62 total REM episodes were obtained (30 of which lasted more than 30 s).

**LFP analysis.** All analysis were performed in MATLAB. For each recording, the position of each recording site on the probe tract (four recording sites per probe) was identified by measuring its impedance while lowering the bundle in saline solution (Na-Cl 0.9%) before implantation. Hippocampal theta and gamma rhythms were confirmed by phase inversion across recording sites in successive hippocampal layers, time-frequency decomposition, and phase-amplitude cross-frequency coupling[37,38]. Although we cannot completely exclude potential contamination of high-frequency LFP recordings by unit activity, we observed that fast gamma oscillations displayed typical phase-amplitude coupling patterns for all recordings robustly across animals. The size of our electrode diameters (25 to 50 microns) and the stability of our recordings throughout sessions decrease the probability that fast gamma events arose from correlated spiking activity. We then selected the putative CA1 and dentate gyrus recording sites based on theta-gamma coupling patterns and computed a differential signal from these two traces for each animal. EEG was first filtered in the LFP range (LFP, 0.1–2 kHz) and band-pass filtered in typical frequency bands including delta (1–4 Hz), theta (6–10 Hz), low-gamma (20–50 Hz), mid-gamma (50–100 Hz), high-gamma (100–150 Hz), and ripple (150–250 Hz). This division has been thoroughly described and proven to be functionally relevant for hippocampal electrographic recordings[38]. The power of LFP oscillations was computed as the square of the raw signal integrated over a sliding Gaussian kernel of a characteristic width of 500 milliseconds to extract its envelope. We also computed time-frequency spectrograms between 1 and 150 Hz using a typical wavelet-based approach (classical wavelet, center frequency 2 Hz, window size 2 s). To account for the attenuation of high frequencies, we multiplied each row of the spectrogram by the value of the corresponding frequency *f*. To select the putative CA1 recording site, we computed the instantaneous theta phase (taking 0° as a theta trough) using a Hilbert transform and derived theta phase time-frequency spectrograms to exhibit phase-amplitude theta-gamma coupling patterns. CA1 recording sites were identified when peak gamma power occurs at the peak of the theta phase, corresponding to the CA1 *stratum lacunosum-moleculare*[37].

**Sleep scoring.** The sleep scoring procedure was based on traditional methods using neck electromyogram, animal movement and LFP to discriminate between wake, NREM sleep and REMS. Thanks to the three-dimensional (3D) accelerometer placed on the head of the animal, we could discriminate between quiet wake (QW) and active wake (AW). Quiet wake was detected when the EMG was high and the animal stood still with its head close to the ground. Active wake was detected when the animal was either moving, walking, running or standing and whisking in the air. When the EMG dropped below a threshold (variable across recordings, set during offline processing) for more than 20 s, we labeled the subsequent period sleep. NREM sleep is characterized by a large amplitude of irregular activity (white noise distribution between 1 and 50 Hz on the time-frequency spectrogram, high ripple activity, and low theta/delta ratio), whereas REMS is

characterized by increased theta and gamma peaks, minimal EMG, and decreased ripple power. Brief awakening robustly followed the REM episode, as shown in Fig. 1.

**Baseline evaluation and spatial averaging**. fUS data have been shown to be proportional to local CBV. Because it is not possible to derive absolute CBV levels, we normalized ultrasound data. We performed voxel-wise normalization by selecting a baseline period, which we chose to be the first 3 min of QW. If our recording lasted more than 1 h, we included three more minutes, repeating this process for each extra recording hour, to account for any potential drift. We extracted the distribution for each voxel during this baseline period and computed a mean value, leading to one value for each voxel of the image. To derive a signal similar to $\Delta F/F$, we subtracted the mean and divided by the mean for each voxel in the film containing our power Doppler images. This allowed normalization and rescaling of ultrasound data, yielding to an expression in terms of the percent of variation relative to baseline. Each voxel was normalized independently before performing spatial averaging. Our dataset consisted of Doppler films sampled at different rates varying from 3 images s$^{-1}$ to 1 image s$^{-1}$. To homogenize it, we resampled (linear interpolation for each voxel) Doppler films to achieve a frame rate of 1 image s$^{-1}$ for all recordings.

**Atlas registration**. To assess inter-individual variability and perform statistical analysis, we segmented each two-dimensional (2D) recording plane into anatomical regions based on a 3D MRI-based whole-brain atlas, which provided labeling for 52 brain regions[48]. To derive the functional regions in our ultrasound image, we designed a customized registration algorithm, which projected our 2D ultrasound plane onto a 3D volumetric dataset. In short, we manually pinpointed landmarks on the ultrasound image including the outer cortex edges, inner cortex edges, midline plane, and dentate gyri edges, which were prominent due to hippocampal longitudinal arteries wrapping them. We defined nine parameters, including three offset values, three scaling values, and three rotations (13 parameters for multiplane registration), to identify a given plane unambiguously in the 3D Waxholm space. We performed optimization using the simplex algorithm to minimize a global error based on the position of our landmarks and the closest corresponding border in the Waxholm space. Provided the algorithm did not start too far from the actual position, it converged quickly and provided robust registration for any ultrasound plane, including diagonal planes. This process allowed us to derive vascular activity in 20 regions that were observed in two imaging planes intersecting the recording electrode tracts (one coronal view at coordinates Bregma = − 4.0 mm and one diagonal plane 45° relative to the sagittal plane to include the ventral and dorsal hippocampus).

**fUS correlation analysis**. We investigated changes in functional connectivity between anatomical regions by measuring pairwise correlated variations in regional CBV signals. Correlation matrices for functional connectivity analysis were obtained by isolating the frames corresponding to each behavioral state (wake, slow-wave sleep, and REMS) and computing the zero-lag correlation between each pair of functional regions. No Fisher transformation or filters were applied. Each recording led to four connectivity matrices corresponding to each behavioral state, and we computed a mean matrix by averaging each of these matrices across all recordings (Fig. 2b). Statistical significance of the Pearson coefficient was assessed using a Student table with n-2 degrees of freedom, $n$ being equal to the number of pairs used to calculate the Pearson coefficient.

**LFP-CBV correlation analysis**. To assess the association between LFP events and CBV variables, we searched for correlations between each possible combination of one LFP band-pass filtered signal and a given regional CBV variable. As neurovascular processes are not instantaneous, we should consider possible delays between electrographic and vascular signals. A first approach that we rejected here was to directly compute lagged cross-correlations between two signals for any LFP-CBV pair and any lag in a given time window. This was tedious and introduced another variable (correlation lag), which made the overall analysis less clear. Instead, we chose to isolate regional peaks in each LFP envelope signal and CBV regional averages by detecting the zero-derivative (keep only the maxima by looking at the sign of the second derivative). This process was performed for each LFP and each CBV variable independently and yielded a series of timings (tpeak) and values (vpeak). For each point in the timing series of CBV variables, we searched a timing window of [tpeak − 4.0 s; tpeak + 1.0 s] for the presence of a peak in the LFP signal. If a peak was found, we noted the delay between tpeak_CBV and tpeak_LFP and paired up corresponding values [vpeak_CBV; vpeak_LFP]. If more than one peak was found in this interval, which scarcely happened, we selected that closest to the vascular peak. If no peak was found, we discarded the corresponding CBV peak from the analysis but kept a record of the proportion of LFP peaks found relative to the CBV peak. We then computed correlations for all recordings using [vpeak_CBV; vpeak_LFP] pairs. To remove spurious effects due to inter-individual averaging (for example, Yule-Simpson effects), we normalized vpeak_CBV and vpeak_LFP time series between 0 and 1, corresponding respectively the minimal and maximal value in each recording. This artificially increased the number of border points. The timing analysis was performed to compare the mean of the LFP-CBV delay distributions (where delay $\Delta t$ = tpeak_CBV − tpeak_LFP) for each LFP

envelope signal. Statistical significance of the Pearson coefficient was assessed using a Student's table with n-2 degrees of freedom, $n$ being equal to the number of pairs used to calculate the Pearson coefficient.

**Detection of phasic episodes**. Vascular surges were detected by segregating each voxel in each REMS frame of the Doppler video between active and inactive states. A voxel was considered active when its value was higher than an activation threshold, which we set as one standard-deviation above the mean of the active wake distribution for this very pixel (Z-score > 1). This led to a threshold image for each recording. To detect phasic vascular activity, we isolated time periods containing more than 50% of brain voxels above this activation threshold. This approach ensured that both sustained activity and the spatial extent were considered to discriminate between phasic and tonic vascular regimes. We also excluded surges that lasted less than 5 s, which occurred scarcely.

**Statistics**. All statistics are given as + /− standard error of the mean unless stated otherwise. Statistics in Fig. 2a are computed on $n = 7$ animals over 30 recording sessions. Distributions and their mean values are compared using Cohen's distance relative to QW and the two-tailed Mann–Whitney test. Confidence intervals of CBV levels in Table 2 are extracted by first computing and mean value of the temporal series of CBV values for each 30 s time window (which results in downsampling the CBV temporal series, in order to avoid artificially narrow confidence intervals due to high temporal sampling rate). The CBV series are then aggregated for each animal and confidence intervals are computed, with resulting $p$-values obtained from Mann–Whitney test, when comparing to QW distribution.

Matrices shown in Fig. 2b are computed by averaging matrices from 17 recordings (coronal planes) on four animals. Confidence intervals for functional coupling correlation coefficients are computed by applying Fisher transform for each recording and computing the 95% confidence interval. This interval is then averaged for all recordings in the same animal and displayed in Supplementary Table 1. Corresponding Z-scores were computed to test the significance of the difference between two Pearson coefficients R_A and R_B. We compute the ratio $\frac{ZA-ZB}{\sqrt{sd}}$, where Z_A and Z_B are the images of R_A and R_B by the Fischer transform and sd is the cumulative variance of the series A and B ($s_d = s_a^2 + s_b^2$). Here, we tested for the difference of coupling between QW distribution and AW, NREM, and REM distributions.

Bar diagrams shown in Fig. 2c are computed by averaging the mean values of 17 recordings on four animals for the coronal planes and 11 recordings in four animals for the diagonal planes. Statistics are computed using a two-tailed Mann–Whitney test. The significance of Pearson correlation coefficients shown in Figs. 3 and 4c are assessed by computing the $t$-value (using $= \frac{r\sqrt{1-r^2}}{\sqrt{n-2}}$) and reporting it in Student's table with $n$–2 degrees of freedom. The histograms presented in Fig. 3e were computed on $n = 7$ animals over 35 REM episodes, which lasted more than 30 s. Statistical significance of the difference in delays was computed using Student's paired $t$-test between each pair of series of delays.

**Code availability**. The code used to generate the results that are reported in this study is available from the corresponding authors upon reasonable request.

## Data availability

All data and software supporting the findings of this study are available from the corresponding authors upon reasonable request.

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

## Acknowledgements

We would like to thank S. Hubatz for help with the track-running protocol, T. Watson for technical help with current-lesioning protocol and A. Dizeux for technical help with supplementary videos. A.B. received funding from the Ecole Doctorale Frontières du Vivant, Program Bettencourt. The research leading to these results has received funding from the European Research Council under the European Union's Seventh Framework Program (*FP7/2007–2013*)/ERC grant agreement no. 339244-FUSIMAGINE. This work was also partly supported by the Fondation pour la Recherche sur le Cerveau (FRC) (Program Rotary–Espoir en tête).

## Author contributions

A.B. designed the recording electrodes and performed the surgeries and electrode implantation, as well as the training and recording sessions. T.D. and C.D. programmed the ultrasound sequences and clutter-rejection algorithms. A.B. and I.C. designed the experiment and analyzed the behavioral and electrographic data. I.C. programmed the acquisition software and atlas-registration algorithm. A.B. and M.T. analyzed the ultrasound data and discussed multimodal analysis. All authors wrote the paper.
