## [Peer Review File · Nature Communications]

Reviewers' Comments:

Reviewer #1:

Remarks to the Author:

This is essentially the first study since that of Reivich et al (1968; ref 16) to evaluate the flow of blood in the brain during sleep states, with a particular emphasis on flow during REM sleep. The early work used the diffusion and trapping of radiolabeled antipyrine by vessel - yet this approach was later criticized by one of the originators of the technique (Eckman et al 1975 Am J Physiol), who noted "... the uptake of [¹⁴C]antipyrine by cerebral tissues is diffusion limited as well as flow limited, and it therefore is not an ideal tracer for the autoradiographic [local cerebral blood flow] technique." In the present work, the authors used Doppler shifts in ultrasound radiation to infer the speed of RBCs to estimate fractional increases in blood flow. This is a welcome revisiting of an important issue as it bears directly of global changes in brain metabolism that may occur during different stages of sleep. The authors find a large increase in flow during REM sleep through all regions of the brain, with the greatest effect in the hippocampus but comparable effects in the cerebral cortex, etc. I am impressed by this work. Unless I have missed some large systematic error, I feel that this manuscript presents careful measurements on an important topic and should be published after the authors address some queries:

The authors technique is highly accurate for speed (presumably! - a bit of insanity is that commercial laser Doppler equipment cuts off the maximum frequency shift so that speeds are underestimated) but may have other issues - such as estimates of volume and biases related to the orientation of vessels relative to the front of the outgoing ultrasonic field - that should be highlighted for the reader. These systematic issues should be discussed in terms of estimates of how they would effect the quantification of increases in flow.

Recent work by Mateo et al (Neuron 2017) showed that fluctuations in gamma power (read gamma bursts) proceed an increase in arteriole diameter by ~ 2 s and by changes in oxygenation by ~3 s. It would be useful for the authors to compute the correlation between the high gamma and the changes in CBV (figure 3a), note the lag time and note the relation of this time to the lag times in the prior study. The authors will need to reduce their window size from 2 s (Methods) to < 1 s.

Reviewer #2:

Remarks to the Author:

The authors perform a very timely study using simultaneous fUS and LFP recordings to study REM sleep. They report a hyperemic state during REM sleep that exceed blood flow during active and quite wake states. These findings offer new information about the REM sleep states in rodents.

Abstract

"lack of multimodal approaches" should be "lack of multimodal recording approaches"

"local hippocampal oscillations" is an ill-defined termed. Especially for the abstract, the description (ripples, theta rhythm) should be more generic. I don't believe the term "local hippocampal ripples" appears anywhere else in the manuscript.

Main text

The description in the first sentence refers to experimental paradigms used in animal models not in humans. This should be clarified.

In this opening paradigm the work on hippocampal replay in rodents (rats) during sleep should be cited as well. It is mentioned later on. There is work by Loren Frank's group at UCSF and by Matt

Wilson's group at MIT that should be mentioned.

Page 3.

I appreciate that the authors want to create a context for their work. However, no one who studies sleep believes that the overall brain states of REM and wake are the same. This statement is quite inaccurate. The neurophysiology of REM sleep, high cholinergic tone in the cortex and paralysis of muscles, irregular heart rate, irregular breathing, rapid movement of the eyes, genital erection (all of these are controlled by the brain either directly or indirectly) are quite different from the wake state. During REM sleep there is still substantial inactivation of many of the brainstem arousal nuclei relative to the wake state when these nuclei are quite active. These facts are well known.

"lack of appropriate techniques". This sounds pejorative. Scientist to date have made the best possible inferences using the techniques they have available. I am sure that improvements can be made upon the techniques presented here.

An important caveat to the work presented here is that sleep in rodents differs dramatically from sleep in humans. The extent to which the current findings can be extrapolated to humans may be quite limited. This should be mentioned in the Discussion. Speculation about how similar work could be done in humans merits a couple of sentences in the Discussion.

Page 5.

The authors should provide a statistical analysis, ideally within animal to show that the CBV differences between different behavioral states are real effects (statistically significant as measured by confidence intervals as opposed to p-values). There should be enough data to do a compelling analysis within animal. The confidence intervals would make apparent that the observations being reported here are weaker than the p-values may suggest.

In the Methods/Statistics section the authors state that 7 animals were studied. How many VS events were there per animal? Were they roughly the same? Did one animal contribute most of them?

Page 5-6.

The analyses of the ratios of increases in functional coupling are reported without any statistical analyses. Again, confidence intervals computed within animal would be preferred to aggregate comparisons of states using p-values.

What does the denominator 611 correspond to? Is this all of the VS events detected across all animals? It is not clear how to interpret the low gamma, mid gamma, high gamma and theta ratios? Is the statement that some type of gamma event preceded a VS event? A high fraction 428/611 had theta events preceding them.

The statement that the LFP mid gamma and high gamma predict CBV intensity is a bit strong. There is substantial variability around the correlation line as shown in the graph. Indeed, the fraction of variation in the CBV amplitude explained by the LFP gamma power is actually $0.7^{*2} = 0.49$.

Discussion

Page 7. It is incorrect to state that sleep physiologists believe that the state of the brain during REM and awake are similar. This is a substantial oversimplification of modern sleep research in both animals as well as in humans.

Page 8.

The REM episodes are a rarity in rodent sleep not in human sleep. This should be clarified.

"We have also demonstrated a robust association between the fast gamma oscillations and whole-brain vascular hyperactivity, ..." This is a very accurate statement of the findings in the current study.

The authors should provide a reference for their conjecture that the fast gamma "oscillations are triggered by direct entorhinal input from the CA1 region."

Page 9.

It has long been appreciated that REM sleep is an active brain state different from sleep. Why should it

have been previously assumed that the energy levels of REM sleep are lower than in wake states? If during sleep, and in particular, during REM sleep brain regions are performing a different function than during the awake state, it makes sense that the energy needs could be at a different, possibly higher, homeostatic set point. I think the authors should make a statement along these lines rather than making it seem like their findings call into question the fundamental way in which sleep physiologists view REM sleep.

Reviewer #3:
None

Reviewer #1 (Remarks to the Author):

This is essentially the first study since that of Reivich et al (1968; ref 16) to evaluate the flow of blood in the brain during sleep states, with a particular emphasis on flow during REM sleep. The early work used the diffusion and trapping of radiolabeled antipyrine by vessel - yet this approach was later criticized by one of the originators of the technique (Eckman et al 1975 Am J Physiol), who noted "... the uptake of [14C] antipyrine by cerebral tissues is diffusion limited as well as flow limited, and it therefore is not an ideal tracer for the autoradiographic [local cerebral blood flow] technique."

- A reference to this publication has been added in the main text [lines 229-231].

In the present work, the authors used Doppler shifts in ultrasound radiation to infer the speed of RBCs to estimate fractional increases in blood flow. This is a welcome revisiting of an important issue as it bears directly of global changes in brain metabolism that may occur during different stages of sleep. The authors find a large increase in flow during REM sleep through all regions of the brain, with the greatest effect in the hippocampus but comparable effects in the cerebral cortex, etc. I am impressed by this work. Unless I have missed some large systematic error, I feel that this manuscript presents careful measurements on an important topic and should be published after the authors address some queries:

- We thank the reviewer for these positive comments.

The authors technique is highly accurate for speed (presumably! - a bit of insanity is that commercial laser Doppler equipment cuts off the maximum frequency shift so that speeds are underestimated) but may have other issues - such as estimates of volume and biases related to the orientation of vessels relative to the front of the outgoing ultrasonic field - that should be highlighted for the reader. These systematic issues should be discussed in terms of estimates of how they would affect the quantification of increases in flow.

- We agree that a more complete description of the signal estimated by ultrafast power Doppler processing should be given for the readers. A detailed paragraph addressing this point and a reference to previous work from our group [Macé et al 2013; Tanter & Fink, 2014; Demené et al. 2016] have been added to the Discussion. [lines 282-290]

Recent work by Mateo et al (Neuron 2017) showed that fluctuations in gamma power (read gamma bursts) proceed an increase in arteriole diameter by ~ 2 s and by changes in oxygenation by ~ 3 s. It would be useful for the authors to compute the correlation between the high gamma and the changes in CBV (figure 3a), note the lag time and note the relation of this time to the lag times in the prior study. The authors will need to reduce their window size from 2 s (Methods) to < 1 s.

- An additional figure [Figure 5] and a full dedicated paragraph in the main text addressing this point have been to the paper. [lines 190-221]

Reviewer #2 (Remarks to the Author):

The authors perform a very timely study using simultaneous fUS and LFP recordings to study REM sleep. They report a hyperemic state during REM sleep that exceed blood flow during active and quite wake states. These findings offer new information about the REM sleep states in rodents.

Abstract

“lack of multimodal approaches” should be “lack of multimodal recording approaches”

- This has been modified in the text [line 21].

“local hippocampal oscillations” is an ill-defined term. Especially for the abstract, the description (ripples, theta rhythm) should be more generic. I don't believe the term “local hippocampal ripples” appears anywhere else in the manuscript.

- This has been modified in the text [line 25-26].

Main text

The description in the first sentence refers to experimental paradigms used in animal models not in humans. This should be clarified.

- This has been clarified in the text [line 31].

In this opening paradigm the work on hippocampal replay in rodents (rats) during sleep should be cited as well. It is mentioned later on. There is work by Loren Frank's group at UCSF and by Matt Wilson's group at MIT that should be mentioned.

- References to the work of Loren Frank's group [Jadhav et al. Science 2012] and Matt Wilson's group [Wilson & McNaughton, Science, 1994; Lee & Wilson, Neuron, 2001] have been added to the text. [lines 34-37]

Page 3. I appreciate that the authors want to create a context for their work. However, no one who studies sleep believes that the overall brain states of REM and wake are the same. This statement is quite inaccurate. The neurophysiology of REM sleep, high cholinergic tone in the cortex and paralysis of muscles, irregular heart rate, irregular breathing, rapid movement of the eyes, genital erection (all of these are controlled by the brain either directly or indirectly) are quite different from the wake state. During REM sleep there is still substantial inactivation of many of the brainstem arousal nuclei relative to the wake state when these nuclei are quite active. These facts are well known.

- We fully agree that we should tone down this assertion and limit the statement to LFP recordings. This has been modified accordingly in the main text. [lines 47-50, 52-53]

“lack of appropriate techniques”. This sounds pejorative. Scientist to date have made the best possible inferences using the techniques they have available. I am sure that improvements can be made upon the techniques presented here.

- This has been modified in the main text. [lines 58-62]

An important caveat to the work presented here is that sleep in rodents differs dramatically from sleep in humans. The extent to which the current findings can be extrapolated to humans may be quite limited. This should be mentioned in the Discussion. Speculation about how similar work could be done in humans merits a couple of sentences in the Discussion.

- A paragraph addressing this important point has been added to the Discussion. [lines 275-282]

Page 5. The authors should provide a statistical analysis, ideally within animal to show that the CBV differences between different behavioral states are real effects (statistically significant as measured by confidence intervals as opposed to p-values). There should be enough data to do a compelling analysis

within animal. The confidence intervals would make apparent that the observations being reported here are weaker than the p-values may suggest.

- This statistical analysis has been done and detailed in Table 2 (formerly Table 1) which has been modified accordingly to make the confidence intervals apparent for each sleep state across animals and in the main text [lines 129-137].

In the Methods/Statistics section the authors state that 7 animals were studied. How many VS events were there per animal? Were they roughly the same? Did one animal contribute most of them?

- An additional table [Table 1] and a paragraph in the main text addressing this point have been to the paper. [lines 108-119]

Page 5-6. The analyses of the ratios of increases in functional coupling are reported without any statistical analyses. Again, confidence intervals computed within animal would be preferred to aggregate comparisons of states using p-values.

- This statistical analysis has been done and detailed in the main text and in Table S3. The details on about confidence intervals are given in the Methods section. [lines 142-147]

What does the denominator 611 correspond to? Is this all of the VS events detected across all animals? It is not clear how to interpret the low gamma, mid gamma, high gamma and theta ratios? Is the statement that some type of gamma event preceded a VS event? A high fraction 428/611 had theta events preceding them.

- An additional figure [Figure 4] and a paragraph in the main text addressing this point have been to the paper. [lines 177-189]

The statement that the LFP mid gamma and high gamma predict CBV intensity is a bit strong. There is substantial variability around the correlation line as shown in the graph. Indeed, the fraction of variation in the CBV amplitude explained by the LFP gamma power is actually $0.7^{*2} = 0.49$.

- This has been modified in the abstract [line 27] and in the legend of Figure 3.

Discussion

Page 7. It is incorrect to state that sleep physiologists believe that the state of the brain during REM and awake are similar. This is a substantial oversimplification of modern sleep research in both animals as well as in humans.

- This has been modified in the text. [lines 223-225; lines 226-228]

Page 8. The REM episodes are a rarity in rodent sleep not in human sleep. This should be clarified.

- This has been modified in the text. [lines 238-240]

“We have also demonstrated a robust association between the fast gamma oscillations and whole-brain vascular hyperactivity, ...” This is a very accurate statement of the findings in the current study.

The authors should provide a reference for their conjecture that the fast gamma “oscillations are triggered by direct entorhinal input from the CA1 region.”

- References to the work of Gyorgyi Buzsaki’s group [Montgomery et al. JNeuro 2008; Belluscio et al. JNeuro 2012; Schomburg et al. Neuron 2014] have been added to the text and the sentence have been rephrased. [lines 245-253]

Page 9. It has long been appreciated that REM sleep is an active brain state different from sleep. Why should it have been previously assumed that the energy levels of REM sleep are lower than in wake states? If during sleep, and in particular, during REM sleep brain regions are performing a different function than during the awake state, it makes sense that the energy needs could be at a different,

possibly higher, homeostatic set point. I think the authors should make a statement along these lines rather than making it seem like their findings call into question the fundamental way in which sleep physiologists view REM sleep.

- This has been modified and a statement in accordance with this point has been included in the main text. [lines 291-294]

Reviewers' Comments:

Reviewer #1:

Remarks to the Author:

The authors have responded fully to both reviewers in terms of new analysis and additions and revisions to the main text. I am quite pleased with the correlation analysis that is now part of Figure 5 and the final paragraph of the RESULTS. I support publication of these unique and important findings and congratulate the authors.

Reviewer #2:

Remarks to the Author:

The authors have successfully addressed all of my concerns.

REVIEWERS' COMMENTS:

Reviewer #1 (Remarks to the Author):

The authors have responded fully to both reviewers in terms of new analysis and additions and revisions to the main text. I am quite pleased with the correlation analysis that is now part of Figure 5 and the final paragraph of the RESULTS. I support publication of these unique and important findings and congratulate the authors.

- We thank the reviewer for these positive comments.

Reviewer #2 (Remarks to the Author):

The authors have successfully addressed all of my concerns.

- We thank the reviewer for these positive comments.